# An Abattoir-Based Study on the Prevalence of *Salmonella* Fecal Carriage and ESBL Related Antimicrobial Resistance from Culled Adult Dairy Cows in Wuhan, China

**DOI:** 10.3390/pathogens9100853

**Published:** 2020-10-19

**Authors:** Jie Wang, Kaili Xue, Ping Yi, Xiaojie Zhu, Qingjie Peng, Zijian Wang, Yongchong Peng, Yingyu Chen, Ian D. Robertson, Xiang Li, Aizhen Guo, Joshua W. Aleri

**Affiliations:** 1The State Key Laboratory of Agricultural Microbiology, Wuhan 430070, China; Jie.Wang@murdoch.edu.au (J.W.); xuekaili1213@163.com (K.X.); 18778080924@163.com (P.Y.); Xiaojie.Zhu@murdoch.edu.au (X.Z.); pengqingjie2007@163.com (Q.P.); captainwang@aliyun.com (Z.W.); pengyongchong@webmail.hzau.edu.cn (Y.P.); chenyingyu@mail.hzau.edu.cn (Y.C.); I.Robertson@murdoch.edu.au (I.D.R.); 2School of Veterinary Medicine, College of Science, Health, Engineering and Education, Murdoch University, 90 South Street, Murdoch 6150, Western Australia, Australia; 3College of Veterinary Medicine, Huazhong Agricultural University, Wuhan 430070, China; 4Hubei International Scientific and Technological Cooperation Base of Veterinary Epidemiology, Huazhong Agricultural University, Wuhan 430070, China; 5Wuhan Keqian Biology Co. Ltd., Wuhan 430070, China; 6College of Animal Science and Technology, Huazhong Agricultural University, Wuhan 430070, China; xxianglli@mail.hzau.edu.cn

**Keywords:** bla_CTX-M_ gene, slaughterhouse surveillance, *S.* Typhimurium

## Abstract

The objective of this study was to estimate the fecal carriage of *Salmonella* spp. among culled adult dairy cows presented to an abattoir in Wuhan, China and to evaluate their antimicrobial resistance profiles. Rectal swabs from 138 culled cows were cultured. Laboratory analysis involved the identification of *Salmonella*, the susceptibility assessment and the presence of Extended Spectrum β-lactamases and *mcr* genes in the isolates. An overall prevalence of *Salmonella* of 29.0% was recorded with 63.4% (26/41) and 2.4% (1/41) of the isolates identified as *S*. Typhimurium and *S*. Dublin, respectively. The occurrence of *Salmonella* was higher (odd ratios: 3.3) in culled cows originating from the northeast zone of China than cows originating from the central and north zones. Twenty multi-drug resistant strains (resistant to three or more antimicrobial agents) were detected (48.8%) and overall, a high resistance to ampicillin (36/41) and tetracycline (15/41) was observed. Extended Spectrum β-lactamases phenotypes were found in 7/41 isolates, of which all contained the bla_CTX-M_ resistance gene, and no *mcr* genes were found by polymerase chain reaction. The high prevalence of *Salmonella* fecal carriage and antimicrobial resistance may contribute to an increased risk of *Salmonella* transmission to food.

## 1. Introduction

Salmonellosis is a bacterial zoonotic disease that affects a wide range of domestic animals, including chickens, pigs, and cattle, as well as humans [1,2,3]. The disease causes fever, diarrhea, dehydration, and even death among vulnerable people such as children, the elderly, and immunocompromised individuals [4,5]. In livestock, the disease manifests through various clinical signs including acute gastroenteritis (especially in young animals), abortion storms, fever, and sepsis [2,6]. In humans, *Salmonella* strains, other than *S.* Typhi and *S*. Paratyphi, are referred to as non-typhoidal *Salmonella* (NTS) and are predominately found in animal reservoirs [3,6,7]. It has been estimated that internationally, 78.7 million foodborne illness cases were caused by NTS in 2010 [8]. In dairy cattle production systems, the serovars of importance are *S*. Typhimurium and *S*. Dublin, which can be transmitted to humans via the food chain [3,6,7].

The dairy industry in China, like in most countries, is faced with increased pressures for intensification in order to meet the growing demand for dairy products and also to be able to operate profitably [9,10]. Intensification increases the risk of outbreaks of infectious diseases, such as salmonellosis, impacting negatively on both animal and human health and welfare [11]. Despite data showing *Salmonella* to be the second most significant bacterial foodborne pathogen in China with a prevalence of 25% in healthy dairy cattle, there is limited surveillance information on its shedding patterns, resistance profiles, and its economic impact [6,12,13]. While the use of antimicrobials plays an important role in the control and treatment of advanced *Salmonella* infections, the emergence of multi-drug resistance (MDR) in this bacterium poses a severe threat globally [11,14]. This resistance has been extensively documented in the poultry and pig industries [2,7,15]; however, there are few reports in cattle production systems in China [13]. Dairy products contaminated by fecal material have been implicated in sporadic cases and outbreaks in humans of non-typhoidal salmonellosis [11]. Surveillance programs involving sampling animals presented to abattoirs are not only beneficial to provide information on the frequency of carriage, animal groups at risk, and antimicrobial resistance profiles of isolates, but can also assist in aiding critical decision making as part of food safety.

Extended-spectrum β-lactamases (ESBLs) are the enzymes responsible for causing resistance to antibiotics such as penicillins and cephalosporins [16]. Globally, the important ESBL genes that have been reported to be responsible for multiple antibiotic resistance include plasmid-mediated bla_CTX-M_, bla_TEM_, bla_SHV_ and bla_KPC_ genes [17]. Recent studies in China have demonstrated a high prevalence of ESBLs in *Escherichia coli* from cattle [18,19,20,21]. There is a need to monitor the presence of ESBL in *Salmonella* in cattle raised under various production systems. Therefore, the objectives of this study were to estimate the occurrence (prevalence) of *Salmonella* in culled adult dairy cows presented to an abattoir in Wuhan, China and to evaluate their antimicrobial resistance profiles as part of a decision support tool investigating the occurrence of *Salmonella* in dairy farms in China. It was hypothesized that the majority of the culled cows would be positive for *Salmonella*.

## 2. Results

### 2.1. General Description

A total of 138 culled adult dairy cows were sampled from the 393 presented to the abattoir during the period of this study. A total of 134 animals were from north, northeast, northwest, east, and central China, with the location of 4 animals not specified.

### 2.2. Occurrence and Prevalence of Salmonella

The animal level prevalence of *Salmonella* was 29.0% (40/138; 95% confidence intervals (CI): 21.6, 37.3), with one cow having two isolates cultured (total 41 isolates). Further characterization revealed 63.4% (26/41; 95% CI: 46.9, 77.9) and 2.4% (1/41; 95% CI: 0.1, 12.9) of the *Salmonella* isolates to be *S*. Typhimurium and *S*. Dublin, respectively. The remaining 14 *Salmonella* isolates could not be characterized further. The prevalence of *Salmonella* in culled cows ≥5 years (32.6%; 95% CI: 23.2, 43.2) was similar to that for younger cows <5 years (21.7%; 95% CI: 10.9, 36.4) (odd ratios (OR): 1.7; 95% CI: 0.8, 4.0).

*Salmonella* was more commonly isolated from the culled cows sourced from the northeast zone (47.5%; 95% CI: 31.5, 63.9) compared to the central (8.5%; 95% CI: 6.3, 38.1) and north (23.1%; 95% CI: 13.5, 35.2) zones (*p* = 0.009) (Table 1).

### 2.3. Antimicrobial Resistance Test

A detailed description of the percentage of the 41 *Salmonella* isolates’ resistance to nine agents is shown in Table 2. Detailed information on the antimicrobial resistance and susceptibility profiles of the 41 isolates can be found in the Appendix A.

Sensitivity towards all the tested antimicrobials was observed only in 2/41 (4.9%, 95% CI: 0.6, 16.5) of the isolates. MDR was recorded in 20 isolates (48.8%, 95% CI: 32.9, 64.9) (Table 1).

### 2.4. ESBL Characterization

ESBL-producing *Salmonella* was found in seven (17.1%; 95% CI: 7.2, 32.1) of the isolates and nine (22.0%; 95% CI: 10.6, 37.6) isolates possessed the bla_CTX-M_ gene. No bla_TEM_, bla_SHV_, bla_KPC_, *mcr*-1, or *mcr*-2 were detected in any of the 41 *Salmonella* isolates. Eight of the nine bla_CTX-M_ positive isolates were cultured from cows sourced from the north and northeast zones of China.

## 3. Discussion

*Salmonella* was cultured from approximately 30% of the sampled culled adult dairy cows at the abattoir in Wuhan, China. The prevalence recorded from this study was slightly higher than that reported in other studies from the USA (25%; 3%) and Italy (0%) in culled dairy cows [22,23,24]. In China, a prevalence of approximately 20% was reported in cattle sampled at multiple abattoirs and it was postulated that the feces and the hides posed a high risk for contamination with *Salmonella* [13]. Yang [25] found 13 *Salmonella* isolates from 78 pieces of beef in marketplaces in Shaanxi and Xu [26] reported the prevalence of *Salmonella* in beef increased recently (9.30–15.79%) in southern China. The high prevalence of *Salmonella* in feces in this research may be related to the place of origin.

With data collected by the National Foodborne Disease Outbreaks Surveillance System, *Salmonella* was reported to be responsible for 13.2% of all human foodborne cases in China in 2013 [12]. The per capita consumption of beef (predominantly sourced from beef cattle, culled dairy cows, and sometimes buffalo) is predicted to increase in China [9], with intensification of domestic production systems and international importation required to meet this demand [27]. Culled dairy cows comprise a large part of beef production in many countries. Currently, for example, in Canada and the USA, approximately 30% of beef production is sourced from culled dairy cows [28,29]. The high prevalence of *Salmonella* detected in the current study is a potential risk to humans, as well as increasing the risk of cross-contamination of carcasses within the abattoir environment [13]. There is a need to determine the occurrence of *Salmonella* within China’s dairy farms and to investigate potential risk factors so that appropriate mitigation strategies can be developed and implemented [13,30,31].

From this study, about 50% of the isolated *Salmonella* demonstrated multi-drug resistance. The recorded multi-drug resistance in this study was higher than an earlier study conducted with cattle in China (0% and 35%, respectively) [6,13] but was lower than that found in dry milk-related infant food (75%) [32]. The antimicrobials with the highest resistance were ampicillin and tetracycline. These antibiotics belonged to the two most commonly used antibiotic groups in 2018 in the Chinese livestock industries, including chicken and pigs [33]. This finding is similar to that reported in culled dairy cows in California [22] and is likely linked to the global long-term use of these antibiotics for promoting growth and controlling mastitis in dairy cows [34].

The findings of this study showed high antimicrobial resistance towards ampicillin (AMP) and tetracyclines (TET). From this study, polymyxin B (PB) and imipenem (IPM) were observed to have the least resistance. These findings are in agreement with which the findings of others [25,35,36]. PB had been used as a therapeutic drug and feed additive since the 1980s in China. Its usage is expected to reach 16 500 tonnes by the year 2021 [37]. IPM, one of the carbapenems, is widely used against multi-resistant Gram-negative pathogens in human medicine, especially strains producing ESBL [38]. The resistance profiles towards PB and IPM are of public health importance as the encoding genes responsible for resistance are located on the mobile genetic elements [37]. Our study showed a low occurrence of the resistance genes (ESBL and *mcr*) in *Salmonella*. The absence of the *mcr* gene in the tested *Salmonella* isolates is likely a result of the ban on the use of polymyxin for promoting the growth of livestock in China in 2016 [39]. The monitoring of resistance genes in bacterial isolates is important to provide evidence on the reduction in the use of antimicrobials [16]. There is a need to determine and assess antimicrobial usage patterns in dairy farms in China, as has been undertaken in other countries [22,40].

In this study, we only collected fecal swabs from culled dairy cows at the abattoir but did not collect samples from other sites, such as the hides, which have been shown to be heavily contaminated with *Salmonella* in another study in China [13]. We were unable to characterize the distribution of *Salmonella* isolates based on the origin source of the cows due to the small sample size. Both the lack of detailed data on farms and of information on transport duration and condition of transportation were further limitations to this study, as others have highlighted the importance of these factors on shedding/carriage of *Salmonella* [41,42]. Our study highlights the potential hazard to food safety arising from *Salmonella* carriage by culled dairy cows processed at an abattoir. There is a need to determine the occurrence and carriage of *Salmonella* among dairy cows in China across the different regions to identify potential risk factors for infection so that appropriate on-farm control measures can be developed.

## 4. Materials and Methods

### 4.1. Study Area and Approvals

This study was conducted in Wuhan, China in accordance with the Australian Code of Practice for the Care and Use of Animals for Scientific Purposes (National Health and Medical Research Council, 2013) and the Guide for the Care and Use of Laboratory Animals (issued by the Committee of Hubei People’s Congress, Hubei, China, 2005) with the approval of the Animal Ethics Committee of Murdoch University (approval number R3201/19) and the Committee on the Ethics of Animal Experiments at Huazhong Agricultural University.

### 4.2. Study Design

This was an abattoir surveillance study conducted from June to July 2019 in Wuhan investigating the prevalence of *Salmonella* in fecal samples collected from adult culled dairy cows from Chinese dairy farms. The selected abattoir processed most of the culled dairy cows for local consumption in Wuhan. The source population included all culled dairy cows at the abattoir during the study time and these originated from the north (Inner Mongolia and Hebei Provinces), northeast (Heilongjiang, Jilin, and Liaoning Provinces), northwest (the Ningxia Hui Autonomous Region), central (Henan and Hubei Provinces), and east (Shandong Province) zones of China (Figure 1).

### 4.3. Sample Size Calculations

Using Epitools (https://epitools.ausvet.com.au) with an assumed prevalence of 50%, 95% confidence intervals, and 10% precision a sample size of 100 animals was deemed appropriate.

### 4.4. Study Populations and Selection of Study Subjects

A total of 138 culled adult dairy cows were sampled in this study. All sampled cows were Holstein Friesians, the predominant dairy breed within the dairy industry in China. A total of 3–5 animals from the 6–10 animals processed each day were selected based on their originating zones. Data on the provinces of origin were recorded prior to sampling. An animal’s age was assessed by examination of their incisors, and then, categorized into four groups (2.5, 3.5, 4.5, and ≥5 years old).

### 4.5. Collection of Fecal Swabs

Feces were collected post stunning by inserting two cotton swabs into the rectum and rotating against the rectal wall. These were then immediately placed into a sterile tube containing 10 mL of buffered peptone water (BPW) (Hopebio Co., Ltd., Qingdao, China), stored on ice, and transported to the State Key Laboratory of Agricultural Microbiology of Huazhong Agricultural University, Wuhan, where they were processed on the day of collection.

### 4.6. Isolation and Identification

The culture and isolation of *Salmonella* on the fecal swabs were conducted according to the international standard methods: EN-ISO 6579:2002/A1: 2007: Amendment 1: Annex D. Briefly, pre-enrichment with BPW was done at 37 ℃ for 18–20 h and thereafter, 100 µl from the pre-enrichment homogenate was inserted on a modified semi-solid Rappaport–Vassiliadis (MSRV) medium (Hopebio Co., Ltd., Qingdao, China), while 1 mL of culture was transferred into 9 mL of Mueller–Kauffmann Tetrathionate novobiocin (MKTTn) broth (Hopebio Co., Ltd., Qingdao, China). Both preparations were then incubated at 41.5 and 37 ℃, respectively, for selective enrichment. After 48 h, suspected growth on the MSRV media and aliquots from all MKTTn Broth were streaked onto Xylose Lysine Deoxycholate (XLD) agar (Hopebio Co., Ltd., Qingdao, China), which was incubated aerobically at 37 °C for 24 h. At least three typical-looking colonies of *Salmonella* on each XLD agar plate were selected and were confirmed by performing a polymerase chain reaction (PCR) with the primers for the *Salmonella* enterotoxin (Stn) gene [43]. After this, *Salmonella* isolates with different colony morphologies but from the same sample were taken and purified for further molecular identification.

### 4.7. Characterization Using Rapid Molecular Detection Methods

The confirmed *Salmonella* were characterized using a PCR according to serotype-specific genes to detect *S.* Typhimurium, *S*. Enteritidis, *S*. Agona, *S*. Dublin, and *S*. Infantis [44,45,46] (Table 3).

### 4.8. Antibiotic Susceptibility Test

Antibiotic susceptibility of the *Salmonella* isolates was determined by the broth microdilution method according to the guidelines recommended by the Clinical and Laboratory Standards Institute (CLSI) [47,48]. A total of 9 antimicrobials representing 9 classes that are commonly used in animals and humans were used. This included AMP, TET, GEN, IPM, and SXT with recommended cut-off values by CLSI [48] and florfenicol (FFC), enrofloxacin (ENR), and ceftiofur (CEF), which had recommended cut-off values by CLSI M31-A3 [47]. The interpreted result of PB was unavailable in the recent CLSI criteria; therefore, a PB ≥ 4 μg/mL, recognized by CLSI in 1981 [49], was considered resistant. AMP, TET, GEN, ENR, FFC, and CEF were selected as these are frequently used in veterinary clinical food animal practice. Susceptibility to PB was also tested, as it can still be used for therapeutic purposes in China, although it has been officially banned for use as a growth-promoter [21]. *Escherichia coli* ATCC 25,922 was used as the quality control strain. Strains resistant to three or more antimicrobial agents were considered as being MDR.

### 4.9. Identification of ESBL-Producing Salmonella

Each *Salmonella* isolate was streaked onto Mueller–Hinton Agar supplemented with 1 µg/mL cefotaxime [50]. Suspected ESBL-producing *Salmonella* on the Mueller–Hinton Agar were further examined by combination disk diffusion according to the CLSI guidelines using cefotaxime (30 μg) ± clavulanic acid (10 μg) and ceftazidime (30 μg) ± clavulanic acid (10 μg) [48]. All isolates were cultured in BPW for 8 h, and then, DNA was extracted from 1mL of the liquid using the boiling method and analyzed with a PCR for bla_CTX-M_ [51], bla_TEM_ [52], bla_SHV_ [53], bla_KPC_ [38], *mcr*-1 [37], and *mcr*-2 [54] genes (Table 4).

### 4.10. Statistical Analysis

Data sorting was done in Microsoft Excel (Microsoft Office 2017) and thereafter analyzed using the statistical software SPSS version 22.0 (SPSS, Inc., Chicago, IL, USA). The prevalence of *Salmonella* at the animal level was estimated. The associations between age groups, locations, and patterns of resistance were determined by calculating odd ratios and their 95% CI. A Chi-square test was used to compare the prevalence between the age groups and locations and the results were interpreted at *p* value ≤ 0.05.

## 5. Conclusions

From this abattoir surveillance, it is concluded that the prevalence of *Salmonella* spp. among adult culled dairy cows was 29%. A high proportion (approximate 50%) of the isolates showed multiple-drug resistance. There is a need to determine the occurrence of *Salmonella* in dairy farms and the potential risk factors, and further develop appropriate on-farm control measures.

## Figures and Tables

**Figure 1 pathogens-09-00853-f001:**
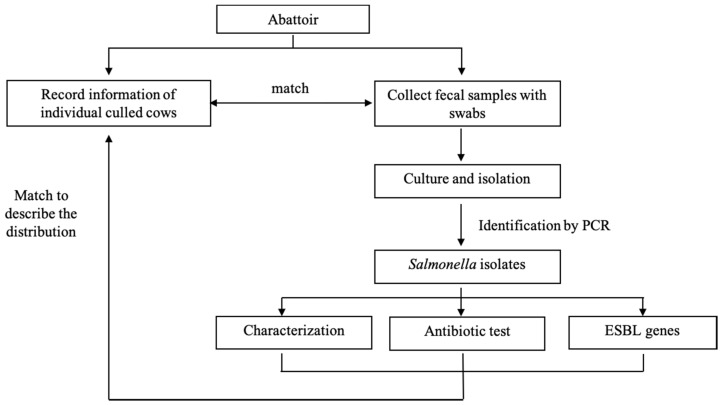
Framework for the assessment of carriage of *Salmonella* in culled cow in abattoir.

**Table 1 pathogens-09-00853-t001:** The place of origin of the culled dairy cows sampled for the prevalence of *Salmonella* and multi-drug resistant (MDR) *Salmonella* (*n* = 40).

Source	Total No. of *Salmonella*-Positive Cases	No. of Culled Dairy Cows Sampled	Prevalence (95% CI)	OR (95% CI)	*p*-Value *	No. of MDR Samples, Percentages (%) 95% CI	No. of *S*. Typhimurium-Positive Samples	No. of MDR of *S.* Typhimurium-Positive Samples, Percentages (%) 95% CI
Central	5	27	18.5% (6.3, 38.1)	1		1, 20.0 ** (0.5, 71.6)	2	0, 0.0 *** (0.0, 84.2)
North	15	65	23.1% (13.5, 35.2)	1.3 (0.4, 4.1)	0.629	8, 53.3 ** (26.6, 78.7)	10	4, 40.0 *** (12.2, 73.8)
Northeast	19	40	47.5% (31.5, 63.9)	4.0 (1.3, 12.6)	0.015	10, 52.6 ** (28.9, 75.6)	13	5, 38.5 *** (13.9, 68.4)
Northwest	0	1	0.0% (0.0, 97.5)	-		-	-	-
East	0	1	0.0% (0.0, 97.5)	-		-	-	-
Unspecified	1	4	25.0% (0.6, 80.6)	1.5 (0.1, 17.2)		1, 100.0 ** (2.5, 100.0)	1	1, 100.0 *** (2.5, 100.0)
Total	40	138	29.0% (21.6, 37.3)	-		10,50.0 (33.8, 66.2)	26	10, 38.5 (20.2, 59.4)

* compared to the central zone. ** Compared with the number of *Salmonella*-positive cows sourced from that zone. *** Divided by the total number of *Salmonella* Typhimurium isolates in each zone.

**Table 2 pathogens-09-00853-t002:** Percentages of the resistance of 41 *Salmonella* spp. isolates to nine tested antibiotics.

Antibiotic Group	Antibiotic	Percentage of Isolates with Resistance (95% CI)
Penicillins	Ampicillin	87.8% (73.8, 95.9)
Tetracyclines	Tetracycline	56.1% (39.7, 71.5)
Cephems	Ceftiofur	39.0% (24.2, 55.5)
Phenicols	Florfenicol	31.7% (18.1, 48.1)
Aminoglycosides	Gentamicin	26.8% (14.2, 42.9)
Quinolones	Enrofloxacin	26.8% (14.2, 42.9)
Folate pathway antagonists	Trimethoprim- Sulfamethoxazole	4.9% (0.6, 16.5)
Polypeptides	Polymyxin	0.0% (0.0, 8.6)
Carbapenems	Imipenem	0.0% (0.0, 8.6)

**Table 3 pathogens-09-00853-t003:** Primers of the *Salmonella* serotype genes used in this study.

Serotype	Sequence (5′-3′)
Dublin	GAGATTGCCGATGCTTTTCC
	AACCTGCTCTACGGGTCTGATT
Agona	AATTGTCTGCGTCATTGAGTTGGA
	CGGCGGTTCTTCATCTATCTTCG
Typhimurium	AAAAGCAGGCATGTCCACCG
	ATCCCGCAGCGTAAAGCAAC
Enteritidis	GCCACTGTCTGATGCTCTTG
	GAAAGGCTCCGTGGTTAGT
Infantis	AGCCAACGCCACCTACTACT
	TGAACACCATATCCATCCACAT

**Table 4 pathogens-09-00853-t004:** Primers of the ESBL- resistant genes used in this study.

Genes	Sequence
bla_CTX-M_	ATGTGCAGYACCAGTAARGTKATGGC
	TGGGTRAARTARGTSACCAGAAYCAGCGG
bla_TEM_	ATGAGTATTCAACATTTCCG
	CTGACAGTTACCAATGCTTA
bla_SHV_	TTATCTCCCTGTTAGCCACC
	GATTTGCTGATTTCGCTCGG
bla_KPC_	TGTCACTGTATCGCCGTC
	CTCAGTGCTCTACAGAAAACC
*mcr*-1	CGGTCAGTCCGTTTGTTC
	CTTGGTCGGTCTGTAGGG
*mcr*-2	TGGTACAGCCCCTTTATT
	GCTTGAGATTGGGTTATGA

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
