# Peer review of "An Abattoir-Based Study on the Prevalence of Salmonella Fecal Carriage and ESBL Related Antimicrobial Resistance from Culled Adult Dairy Cows in Wuhan, China"

_pathogens, 2020, doi:10.3390/pathogens9100853_

Round 1
Reviewer 1 Report
The article is well writen with clear aims and conclusions. Presentation of data is clear as well and shown in separate tables. However, there are few questions, some maybe hypothetical.
As authors indicate few limiations can be seen in the work, such as no information about influence of transportation on bacteri spread. How authors think, this additional factors could affect final results? Is it possible that it would have some impact? Still there is a big discrepancy in the number of affected animlas origniating from North and other regions. Could infection occur after leaving the original farm?
Most of the methodology, primers are cited properly but according to authors would it be fine to include separate table with used primers in the methods section?
Results looks fine, however graphical representation is somehow missing in the manuscript. This would be nice to have. Is it possible that authors could include graphical abstract, or possible future methodology for studying this subject in the form of scheme? Are any gene PCR data available to be included, would they be of additional benefit to the paper?
I do not find major issues in the article.
Reviewer 2 Report
Reviewers’ Comments and Authors Response
Paper number: pathogens-959928
Paper title: “An abattoir-based study on the prevalence of Salmonella infection and ESBL related antimicrobial resistance from cows culled adult dairy cows in Wuhan, China”.
I here summarize some minor points:
Abstract:
- Delete acronyms, they are not necessary (MIC, CI….). And “Extended Spectrum β-lactamases” for ESBL.
- Keywords: some names appear in the title: Antimicrobial resistance; dairy cows
Material & methods section: Were some of sampled animals sacrificed for humanitarian reasons? (e.g. “urgence sacrifice”).
Provided the above questions are answered and problems are fixed, the paper can be reconsidered for publication.
I hope the comments below will help you to clarify the manuscript.
Reviewer 3 Report
General
Straightforward, not new, just basic epidemiological data, using established methods.
Overall well written, easy to read, but there are numerous small grammatical errors which would benefit from editorial help; e.g. in the sentence ‘the enzymes responsible for causing resistance towards penicillins and cephalosporins antibiotics’, ...it should be ‘the enzymes responsible for causing resistance to penicillin and cephalosporin antibiotics’.
Change ‘cull’ to ‘cullled’ throughout the text.
You did not test for ‘infection’ – you tested for ‘faecal carriage’; correct this throughout. Note you didn't explain why the cows were culled, which should be added in, as it may affect the data.
Methods
Delete the co-ordinates ((29°58′—31°22′N, 113°41′—115°05′E)).
Section 4.8 – This sentence is the wrong way around... ‘PB was tested for its ongoing resistance to Salmonella.’ ... PB is not resistant to Salmonella, it is the Salmonella that is resistant, or susceptible, to PB.
I suggest you re-model the sentence to ... “Susceptibility to PB was also tested, as it can still be used for therapeutic purposes in China, although it has officially been banned for use as a growth-promoter”.
Abstract
You could re-arrange the text as follows, to include more fact, in fewer words. ... “This study aimed to estimate faecal carriage of Salmonella spp. among culled adult dairy cows presented at an abattoir in Wuhan, China, and to evaluate their antimicrobial susceptibility profiles. Rectal swabs from 138 culled cows were cultured; PCR methods were used to identify and serotype suspect colonies. Susceptibility testing followed CLSI guidelines, including phenotypic detection of ESBLs. PCR was used to characterise ESBL and MCR genes. The overall prevalence ...... ”
You don’t mention MCR results in the abstract; put them in.
It may not be appropriate to report ‘MDR’ in the abstract unless you define it in the abstract.
In fact, I can’t see where you define MDR in the main text – please make sure it is in there somewhere!
The last sentence of the abstract is boring. I suggest you delete it, and use the space to compare your results with the international literature, so the reader gets a sense of whether this data is similar, or very different, from the literature.
Introduction
Why did you hypothesize that the majority of the culled cows would be positive for Salmonella, when you reference data showing the prevalence in China is much lower, at 25%?
Results
In table 2, the text says ‘Antimicrobial resistance was highest for ampicillin and tetracycline at 87.8 and 56.1%, respectively.’ .... but this is redundant, as the numbers are in the table! Delete this text.
Discussion
You don’t discuss why the a) carriage rates, and b) resistance rates, were so different in different geographical locations. What differences in animal care might explain this? Do you have data on antibiotic utilisation rates, by province? Are there differences in animal husbandry?
The MCR detection rate was zero, and you mention that this may be because colistin was banned as a growth promoter, but you don't give Salmonella MCR rates from previous years, so the reader doesn't know whether there has been a change or not.
Reviewer 4 Report
This manuscript provides information on Salmonella isolates obtained from culled dairy cows feces in an abattoir form Wuhan. Prevalence data in relation to place of origin, phenotipic antimicrobial resistance and Identification of ESBL-producing Salmonella are shown.
The MS provides original data however, the presentation of the results, especially in relation to the virulence profiles, and the discussion must be improved.
Introduction
-Authors have as an objective “to evaluate their antimicrobial resistance profiles” but profiles of the isolates are not shown.
Results
-The presentation of the complete data, with the details of the characterization of each of the isolates (origin, serotype, antimicrobial resistance profile, ESBL-producing isolate or not), should be available to the readers, for example in a table of the supplementary material.
-In general, antimicrobial data results are not clearly presented. The profiles of the 20 MDR isolates are not available. It is important to know the combination of antimicrobial resistances and their distribution among the isolates. Therefore, these data should be displayed.
Page 3, 2.3 Antimicrobial resistance antibiogram. Change antibiogram by other word. Perhaps, test?
Authors state: “A detailed description of the antimicrobial resistance and susceptibility profiles of the 41 Salmonella isolates is shown in Table 2”. But, in the table 2 susceptibility profiles are not detailed, instead only percentage of isolates with resistance are shown.
Table 2.
The legend “Resistance of 41 Salmonella spp. isolates to each tested antibiotic” is not right. In the table percentages are shown, not resistance of each isolate and each antibiotic.
In the footnote it is stated: “Susceptibility towards all of the tested antimicrobials was observed only in 2/41 (4.9%, 95% CI: 0.6, 16.5) of the isolates.” However, in the table there are not isolates resistant to Polymyxin and Imipenem.
The sentence “Antimicrobial resistance was highest for ampicillin and tetracycline at 87.8 and 56.1%, respectively” can be deleted from the legend.
Table 3. Perhaps, it is not necessary. Its information could be incorporated as two columns in table 1:
All MDR Salmonella / Salmonella Typhimurium
No. of MDR samples (percentage) / No. of MDR samples (percentage
Discussion
This section should be deepened and it would be convenient to discuss previous related works especially from China, some cited by the authors and others, such as for example:
Paudyal N, Hang Pan, Mohammed Elbediwi, Xiao Zhou, Xianqi Peng, Xiaoliang Li, Weihuan Fang, Min Yue. Characterization of Salmonella Dublin isolated from bovine and human hosts. BMC Microbiol. 2019; 19: 226.
Xu Z, Wang M, Zhou C, Gu G, Liang J, Hou X, Wang M, Wei P. Prevalence and antimicrobial resistance of retail-meat-borne Salmonella in southern China during the years 2009-2016: The diversity of contamination and the resistance evolution of multidrug-resistant isolates. Int J Food Microbiol. 2020
Yang B, Qu D, Zhang X, Shen J, Cui S, Shi Y, Xi M, Sheng M, Zhi S, Meng J. Prevalence and characterization of Salmonella serovars in retail meats of marketplace in Shaanxi, China. Int J Food Microbiol. 2010 Jun 30;141(1-2):63-72.
Zhang L, Fu Y, Xiong Z, Ma Y, Wei Y, Qu X, Zhang H, Zhang J, Liao M. Highly Prevalent Multidrug-Resistant Salmonella From Chicken and Pork Meat at Retail Markets in Guangdong, China. Front Microbiol. 2018 Sep 10;9:2104. doi: 10.3389/fmicb.2018.02104. eCollection 2018.
Zhou M, Li X, Hou W, Wang H, Paoli GC, Shi X.
Incidence and Characterization of Salmonella Isolates From Raw Meat Products Sold at Small Markets in Hubei Province, China. Front Microbiol. 2019 Oct 4;10:2265.
